# Sim2Real: Generative AI to Enhance Photorealism through Domain Transfer with GAN and Seven-Chanel-360°-Paired-Images Dataset

**DOI:** 10.3390/s24010094

**Published:** 2023-12-23

**Authors:** Marc Bresson, Yang Xing, Weisi Guo

**Affiliations:** School of Aerospace, Transport and Manufacturing, Cranfield University, Bedfordshire MK43 0AL, UK; weisi.guo@cranfield.ac.uk

**Keywords:** generative AI, GAN, simulation, domain transfer, 360° images, perspective augmentation, photorealism, dataset

## Abstract

This work aims at providing a solution to data scarcity by allowing end users to generate new images while carefully controlling building shapes and environments. While Generative Adversarial Networks (GANs) are the most common network type for image generation tasks, recent studies have only focused on RGB-to-RGB domain transfer tasks. This study utilises a state-of-the-art GAN network for domain transfer that effectively transforms a multi-channel image from a 3D scene into a photorealistic image. It relies on a custom dataset that pairs 360° images from a simulated domain with corresponding 360° street views. The simulated domain includes depth, segmentation map, and surface normal (stored in seven-channel images), while the target domain is composed of photos from Paris. Samples come in pairs thanks to careful virtual camera positioning. To enhance the simulated images into photorealistic views, the generator is designed to preserve semantic information throughout the layers. The study concludes with photorealistic-generated samples from the city of Paris, along with strategies to further refine model performance. The output samples are realistic enough to be used to train and improve future AI models.

## 1. Introduction

This study tackles the scarcity of comprehensive city datasets by creating a Generative Adversarial Network (GAN) capable of generating photorealistic images of a city environment. It does that thanks to a specialised dataset that matches street view photos with corresponding seven-channel simulated images. This approach allows the creation/reproduction of scenarios useful for military purposes, with the need to explore numerous contexts, or autonomous driving with consideration for complex road geometries as they pose problems with current technologies [1].

This study explores the effectiveness of GANs for domain transfer, leveraging the custom dataset as a foundational resource. In this dataset, the target domain is composed of authentic 360° photos captured in the city of Paris by crowdsourcing, serving as the ground truth for the network training phase. In contrast, the simulated domain encapsulates aspects of a three-dimensional scene, such as depth, colour, and surface normal (their role is described later). All that information is packed into 360° seven-channel images to match the form of the target domain. This research adopts a paired image strategy, avoiding reliance on pseudo-ground truth samples. It requires a more rigorous process to position and rotate a virtual camera to have the simulated sample taken from the exact same point of view as the authentic photos. Doing so optimises computational resources, ensuring efficiency to achieve desired performance levels. The 360° images cannot be directly fed into the model as they are too large and impractical, so a transformation that “cuts in” the image is used, along with basic data augmentation steps. Then, to elevate simulated images to photorealistic standards, cutting-edge GAN networks are employed, generating realistic images from a texture-less 3D environment. The generator network, designed to preserve semantic information across layers, exhibits great performance, particularly when uniform inputs are provided, as is often the case in basic 3D representation. To optimise and evaluate the outputs of the model, different losses and metrics were tested and compared.

Throughout this paper, we dive into the challenges posed by the creation and utilisation of custom datasets, along with the benefits of working with 360° images. We discuss GAN architectures and their fitting for that specialised task, providing valuable insights into deep learning model training. After presenting generated results using different methods, this research highlights strategies to further refine model performance, offering a comprehensive understanding of the potential and nuances.

### Related Works

There are a multitude of ways to synthesise images such as fully convolutional networks [2], variational auto-encoder (VAE) [3,4], generative AI using GANs [5,6,7], or diffusion networks [8,9]. However, the usage of conditional GANs [10] for image-to-image translation spread out to become one of the premium choices [11,12,13,14,15]. In such models, two different networks compete against each other. The generator creates realistic images that are indistinguishable from real images by the discriminator. Their popularity comes from the adversarial loss, which serves as the mechanism for outsmarting the other network, thereby achieving more realistic results.

The image-to-image translation conditional GAN can be used for different tasks, such as image segmentation [13,16,17], image inpainting [14,18,19], style transfer [20,21,22,23], or image super-resolution [24]. Their underlying networks often have the same architecture, such as a U-net generator, first introduced by Ronnenberg O. [25] for a segmentation task. It is an encoder–decoder that features a novel way to cope with the disappearance of high-frequency details in the down-sampling part with long skip connections. It concatenates the encoder’s feature maps to the input of the same-level layers on the decoder side. This enables the decoder to access both low-level and high-level features, allowing for more accurate segmentation and improved localisation. Still using skip connections in ResNet blocks, Park T. et al. [26] built the SPADE generator that goes away with the encoder and that effectively generates realistic outputs even when uniform inputs are presented. The generator does not feature an encoder and works either with a variational auto-encoder to provide an initial feature vector or with a subsampled map of the input. In the latter case, the network will be deterministic. Methods to split the workload onto two networks were used to enable fast-inference models and high-fidelity results, such as in [27], with their coarse-to-fine architecture. This two-stage network enables greater fidelity at the cost of lower performances.

When introduced in [5], the GAN discriminator featured a single-scalar prediction to characterise the output. This simple discrimination technique was also used in [12,13], but [22] worked with a more precise discriminator that was able to discriminate each part of an image independently. This so-called PatchGAN [28]—a fully convolutional network—predicts if a crop (or a patch) of the image is real or fake. This prediction is performed throughout the whole image convolutionally, giving a non-binary prediction 2D vector. Lastly, Wang T.C. et al. [29] and Cheng C. et al. [19] built on PatchGAN to create multi-scale discrimination. It was particularly suited for large images and works by having multiple identical discriminators that are fed the same input with a different subsampled ratio. Moreover, [19] used a multi-head attention mechanism [30] to have better context-aware generation. Finally, [31] used multiple different local discriminators that look at different features of the generated image, creating specialised discriminators that perform better all together to reproduce the distribution of photos.

However, all the aforementioned studies focus on three-channel RGB images. The work of Harris-Dewey J. and Klein R. [28] uses higher channel images in a GAN, using Blender (open-source 3D software available for free at www.blender.org, accessed on 10 May 2023) to simulate multiple layers called passes. These layers represent different physical properties of a scene, such as the depth, surface normal, colour of the elements, or illumination by artificial light. The physical information provides further context for the GAN to better interpret the scene and produces realistic global illuminations.

Lastly, the question of ground-truth imagery was raised. Recent works showed that pseudo-ground truth was enough, as in CycleGAN [32] or MUNIT [32]. In such cases, the input and ground truth do not need to match and must only belong to their respective domains. CycleGAN will minimise the cycle error, that is, going from the first domain to the second domain and back to the first. MUNIT prefers to create a single latent space that contains the inputs and the ground truths. Hao Z. et al. [33] trained a separate model to generate the pseudo-ground truth samples that would be later reused to generate video-consistent images with a SPADE network.

Building on the Harris-Dewey J. and Klein R. sample generation methods in Blender, the truth samples used in this study were found on Mapillary’s open API, where street views are freely available and crowdsourced.

The final results presented are realistic enough that the city of Paris can be recognised in validation samples. Nevertheless, artefacts can give away the fact that they were generated by an IA, which—in the current climate of misleading content—can be welcomed.

## 2. Materials and Methods

### 2.1. Data Preparation

#### 2.1.1. Dataset Overview

The dataset is composed of a file listing every image ID and data, and two folders with the street views and their simulated counterparts. It includes approximately 22.000 samples and weighs over 22 GB. It is available to download at https://www.kaggle.com/datasets/marcbresson/paired-street-view-and-simulated-view-in-paris, accessed on 25 September 2023, and the methodology to recreate similar samples for another city (or in a larger number) is described on this study’s GitHub repository.

This dataset has three main characteristics. First, it uses 360° images for both the simulated domain and the target domain. It is the most common format for street views and autonomous driving. Through the mathematical operations described in Section 2.1.3, we can convert a single 360° image into multiple independent perspective images. It is the main source of data augmentation. Then, both the simulated and target domains match perfectly. That means that for one street view, we will have a simulated sample from the exact point of view. A pixel at a specific coordinate will represent the same thing in the two domains. Lastly, the simulated domain contains a lot of information about the city. Notably, we have:The depth represents how far something is from the camera.A colour that hints towards the role something plays in the landscape, such as a street, footpath, park, or river.Surface normal, which represents what direction something is facing.

#### 2.1.2. Data Acquisition

All the results presented in this paper come from data queried over the Parisian region. Paris was chosen for its numerous street views in Mapillary over multiple seasons, weather conditions, and traffic conditions. OpenStreetMap also has 3D building data for Paris, which allows for the generation of the simulated samples in Blender, an open-source 3D software.

##### Mapillary

Mapillary is a free, open-source project supervised by Meta. Their data source comes from crowdfunding, where people voluntarily upload image sequences that are geotagged. They give you access to a free API that allows you to query data for any region. A region consists of tiles with x, y, and z coordinates, and querying street view data is only possible on a specific zoom level (level 14), from which we get a list of street views. From there, we can work our way to the image, along with its location and estimated camera rotation. Mapillary does not use the camera orientation data, and it recomputes the rotation from the raw camera data and the image output.

##### Blender

It is possible to import entire cities in 3D in Blender thanks to Open-Street Map, a crowdfunded and open-source mapping initiative. This initiative includes most of the buildings in the most popular cities, such as Paris. However, buildings are not necessarily well represented, which can be a challenge when a city presents a lot of complex architecture. Once the city is reproduced in the software, street-view counterparts can be generated. It was particularly hard to apply the correct camera rotation because a lot of 3D rotation formalisms exist. Unfortunately, neither Mapillary’s documentation nor the team specified which one was used. Another Meta’s project [34,35] hinted towards a modified version of the axis-angle rotation vector. It is a three-channel vector, where its norm gives the amount of rotation around an axis given by the vector’s direction. This discovery came after months of using a sub-optimal version of rotation, making the virtual camera correctly aligned only under specific conditions. The new rotation method largely contributed to the increased photorealism of the generated samples.

To convert a 3D scene to an image, we need render engines that interpret a scene to output an image. To achieve this task, they have access to a lot of different types of information (called passes) to generate the desired image. The “Cycle” render engine that comes with Blender and supports 360° images gives 30 passes, notably the depth, the surface normal, and the diffused colour, as shown in Figure 1.

Each of the three passes has its own specificities that need careful attention.

The “depth” pass has only one unbounded channel, between 0 and +infinity. The value of each pixel is expressed in metres and rarely goes over 600, except for the sky. Indeed, as it is infinitely far away, the pixel representing the sky had an infinite value that could be reset to 0 m to avoid compressing the statistical distribution. After this step, the channel is normalised between 0 and 1 to keep the values on the same scale as the two other passes.

The “normal” pass has three channels, with values ranging from −1 to 1. It represents how much the face is colinear with each of the corresponding axis (x, y, and z). The values are independent of the camera’s rotation. It can be seen as an answer to the question, “how much is that thing facing North/West/Up?”. This answer will be between −1 (it is facing the opposite way) and 1. For Figure 1, the values were remapped between 0 and 1, explaining the grey sky.

Finally, the diffused colour, shortened to “DiffCol” in Blender, represents the colour of the surface as defined by the Open Street Map. The ground, walls, footpath, water, etc. all have different colours, which give information analogous to a segmentation map.

#### 2.1.3. Data Processing and Data Augmentation

Directly learning the mapping between 360° images was out of the question because it would require too many computational resources, and 360° images are not common enough. The choice of using them as samples was justified by the fact that we can “move around” in such images, hence providing an important data augmentation step. A single 360° can give six entirely different perspective images.

The data processing and augmentation consist of four main steps:Depth channel remapping from 0 to 1.360° equirectangular to perspective transformation.Resizing to a fixed size of 256×256 px.Random horizontal flip.

##### Perspective Transformation

360° images can be tricky to work with as they exist in multiple forms. The most common one is the equirectangular projection. It keeps vertical lines straight but deforms horizontal lines around the horizon, as shown in Figure 2.

Using algorithms, it is possible to extract perspective images (the ones we are most used to) from an equirectangular image given a yaw, pitch, roll, and field of view. The algorithm used in this work was modified from Ting-I Hsieh’s work [36] to refine and optimise the method thanks to hardware acceleration. The idea is to create a map that indicates where to find a pixel value in the original equirectangular image given an x, y coordinate point in the perspective image. Hardware acceleration of matrix calculus lowers the processing time by up to 5 times.

Strangely enough, this process can corrupt the image by placing NaN (not a number) values for some pixels. It appears to happen only on specific samples, with specific transformation parameters, and in a random manner. Running a loop of the same sample transformation would corrupt the output about 5 times out of 200, even when using non-deterministic Pytorch functions on the CPU. The issue was dealt with by detecting when corruption happens and rerunning the transformation with new parameters.

Not all viewpoints in the 360° image were useful, as looking down would not provide great samples because we often see what the 360° camera was attached to (a cyclist’s helmet, a car’s roof, etc.). For this reason, parameters were chosen as follows: the yaw evolves in its full range, from 0 to 360°; the pitch stays between 0° and 60° to avoid only having the sky or the camera mount; the field of view (FOV) ranges between 60° and 120°, which avoids losing too much quality for shallow FOV, below 60°, and avoids having too much distortion when the FOV is too large.

##### Random Transformation

There are multiple ways to perform data augmentation on the original dataset. It can be randomly transformed as the training goes on, or it can be pre-transformed in a step that happens before the training phase. The first method comes with a few advantages: the data richness is greater as the same initial sample will be transformed differently for each epoch, and it does not require huge disc space to store augmented samples.

#### 2.1.4. Data Loading

One of the longest parts of the training phase was loading the samples from the disc. This step includes loading and decoding the ground truth image and loading and decompressing the binary simulated images. This bottleneck was mitigated using Pytorch 2.0.1 a machine learning Python library. It provides quick access to multiprocessing through its ‘Dataloader’ class, where it is possible to offload any data loading or processing without the Python 3.11 global lock interpreter impairing performance.

The initial idea was to offload all the transformations onto four processes (as many as there were cores on the available supercomputer), with hardware acceleration enabled. As it turns out, hardware acceleration in multiprocessing is slower than running the transformations on the CPU. It probably comes from the GPU being fully solicited by the forward step and backpropagation of the model. Eventually, two different transformation steps were involved: one that transformed a sample in the data loading step using multiprocessing, and the other that transformed the whole batch at once in the main thread, actually used to put it on the right computational device.

##### Data Splitting

The 22.000 samples were split into three sub-datasets: one for training containing most of the samples (80%), another to evaluate the model against new samples (20%), and a last one containing only 20 never-seen samples for visualisation purposes. Cross-validation could not be used because of the limited computational resources available.

##### Batch Size

Due to computation limitations, both in terms of time and power, the batch size was optimised to minimise the training time. It all started with the realisation that loading 8 samples was twice as long as loading 4 and that the training process benefits from using larger batches. A small training task was profiled against different batch sizes. The loading times and model training times were recorded and visualised in Figure 3. It highlights that in a single process, the model will have to wait for new samples to be loaded when the batch size is greater than 2. To mitigate this, employing additional processes in the ‘Dataloader’ PyTorch class can reduce the wait time for new samples by the main process. This graphical method can also be used to find the minimal batch size that will minimise the overall running time. Indeed, we can see in Figure 3 that a batch size of 4 is the smallest value for which there is no waiting time with four data loader processes. Nevertheless, a small batch size can lead to noisier gradients when backpropagating, which can lead to more unstable training. Conversely, this excess noise can be great for exploration diversity as more regions of the latent space are explored.

### 2.2. Network Architectures

This work started around the code source of the infraGAN [37] project, which itself is forked from CycleGAN [24] and pix2pixHD [29]. It comes with multiple models integrated, such as a ResNet generator, a U-net generator, a PatchGAN discriminator, and a PixelDiscriminator that gives a pixel-level prediction. Unfortunately, the code base did not provide a usable Python API and was not adapted to the specificity of this study. It had to be almost entirely rewritten to accommodate a documented Python API, along with extensive logging and performance measurements.

#### 2.2.1. U-Net Generator

The first model tested in this study was the U-net, which is a fully convolutional network. Because of the depth of this architecture, the model is helped by batch normalisation which is a technique to help with the stability and convergence of the network. Batch normalisation normalises the activations in each layer by subtracting the mean and dividing by the standard deviation of the mini-batch. By doing so, it helps maintain activations within a reasonable range, preventing them from becoming too large or too small and ultimately allowing for a more stable gradient during backpropagation. The U-net also uses skip connections, as they are particularly useful to preserve fine-grained spatial information throughout the layers. The encoder extracts high-level information, and the decoder, with the help of skip connections, refines the segmentation map while considering details from multiple scales.

#### 2.2.2. Spade Generator

The spade generator [17,26] focuses on generating realistic and high-quality images by effectively incorporating semantic information into the image synthesis process. Like the U-net, the SPADE generator also uses skip connections in ResNet-inspired blocks to capture both local and global contextual information. It introduces two blocks presented in Figure 4 that constitute the generator shown in Figure 5.

We can see the repetitive use of the input segmentation map at every stage of the network. In this study, the segmentation map is the 7-channel simulated image and is resized to match the current convolutional layer dimension. The consistent refeeding of the segmentation map into the network is the main explanation for the photorealism this model can achieve. To compensate for the richer inputs, the network was used with the option “most” and with 128 base filters (the number of filters in the last dark-blue SRB block in Figure 5).

#### 2.2.3. PatchGAN Discriminator

PatchGAN is a small, fully convolutional network that aims at identifying generated samples among the ground truth samples. As Isola P. et al. [22] state, “This discriminator tries to classify if each N × N patch in an image is real or fake”. It runs convolutionally across each image of the batch, giving a non-binary output that is the average of all the predictions, which helps fine-tune the generator to better correct zones where it makes a lot of errors. Its architecture is represented in Figure 6.

Given the small images being used (256×256 px) in this study, a multiscale discriminator as presented by WANG T-C et al. [29] was not implemented.

### 2.3. Loss Functions and Evaluation

#### 2.3.1. Binar Cross-Entropy Adversarial Loss

The adversarial loss term, often referred to as “GAN loss”, is a key component of GANs. It is used to train the generator to produce realistic and convincing outputs that are indistinguishable from real data. The adversarial loss is calculated based on the difference between the discriminator’s predictions for real street views and the generator’s generated street views. It essentially quantifies how well the generator is fooling the discriminator.

As suggested in [22,23,28], the adversarial loss term here is the binary-cross-entropy combined with a sigmoid layer, and it is called “BCEWithLogitsLoss”. It takes the raw output, called logits (so the output before the activation function), of the discriminator and applies the BCEWithLogitsLoss with a computation trick that is more numerically stable than applying the sigmoid activation, and then uses a separate binary cross-entropy loss function. Its expression for one sample of a batch, with *x* as the input and *y* as the target, is the following:$$l\left(x,\;y\right)=y\cdot \log\sigma \left(x\right)+\left(1-y\right)\cdot log\left(1-\sigma \left(x\right)\right)$$

The average of l over each sample is taken to compute the entire batch loss.

#### 2.3.2. L1 Generator Loss

This loss measures how far from the street view the generated image is. L1 loss measures the mean absolute error for each element in *x* and *y*. It is expressed for a single sample as:$$l(x,\;y)=\left|x-y\right|$$

When computed for a batch, it is reduced by the mean of the adversarial loss.

L1 loss encourages the generated images to be closer to the ground truth images in terms of pixel intensity values. This tends to preserve fine details and textures in the generated images, resulting in sharper and more realistic outputs. Moreover, ref. [22] demonstrated that the L1 loss penalises larger deviations between the generated and target images linearly, compared to the quadratic penalty imposed by the L2 loss (mean squared error). This characteristic helps in preventing the blurring effect that can occur with L2 loss, where slight errors in pixel values are magnified in the loss calculation.

#### 2.3.3. Cycle Loss

In the first dataset version, the truth sample was not well aligned with the simulated sample due to the virtual camera being rotated only around one axis. Training a GAN on these low-quality samples could have resulted in artefacts, especially around the buildings’ edges. For this reason, the cycle loss was implemented to counteract the misalignment, as it only required ground truth that belongs to the target domain without the necessity for matching pairs. The key benefit of CycleGAN is the introduction of cycle consistency loss. This loss ensures that if you translate an image from domain A to domain B and then back from domain B to domain A, you end up with an image that is similar to the original. However, the main drawback of this method is the computational resources needed. Indeed, as the CycleGAN handles both ways of translation, you need twice as much memory as with a GAN. If it was not an issue on the lighter networks, it caused crashes on the NVIDIA V100 when the biggest SPADE generators were loaded.

#### 2.3.4. Structural Similarity Index Measure (SSIM)

The SSIM is well suited for realistic image scoring, as it compares three main components of image quality: luminance, contrast, and structure.

○Luminance. It measures the similarity of the luminance (brightness) values between the corresponding pixels in the two images.○Contrast. It evaluates how well the contrast in the two images matches. It considers the local variations in contrast and aims to capture the relationship between pixel intensities.○Structure. It assesses the structural similarity between the images by considering the correlations between neighbouring pixels. It reflects the spatial arrangement of features in the images.

#### 2.3.5. Human Assessment

Another possible evaluation is the human evaluation, often crowdsourced, as in [38] on online platforms. Users are presented with multiple images and are asked to choose which one they prefer. Because generated samples present artefacts that are not always eliminated by loss functions, it is employed to compare the results with other AI models.

## 3. Results

The tests were all run on a single V100 NVIDIA GPU (manufactured by TSMC), along with an Intel Xeon processor (manufactured by Intel) with 32 GB of RAM in total. Each run was limited to 120 h, limiting the training to five days. For context, the SPADE paper had a 5-day training on eight V100 GPUs, with four million samples and a batch size of 32.

Over 120 h, the model trained on more than 6 million samples (15 samples per second). Inference times were three times lower at just over 40 samples per second. When only using a laptop CPU for inference (without hardware acceleration), that rate went down to three samples per second.

### 3.1. Using the SPADE Generator

This test was run for 330 epochs with a batch size of 2, determined with the graphical method described in Section 2.1.4. The test ran smoothly but was cut short by the time limit. The losses were still steadily decreasing on the 330th epoch, as Figure 7 reveals, so the model could have benefited from a longer training time. At this stage, the network did not present any sign of overfitting or underfitting.

However, Figure 8 reveals that the discriminator had trouble fighting against the adversarial loss. The more epochs went by, the less accurate it was. The two validation lines for real and generated samples highlight that it struggles, particularly with generated samples, but tends to have better accuracy when the input is a real street view. The generator may learn too fast compared to the discriminator, which could benefit from a higher learning rate or two training steps for each generator training step.

Finally, the human assertion you can make in Figure 9 shows that the last two samples are quite realistic. They are coherent and show windows, trees, and even storefronts. We can also notice that they differ a bit in the style of the buildings. With the current model, there is no possibility to steer the generated style towards the style of a given image. It is possible to adapt the SPADE generator to also use a style vector alongside the input. That would allow the user to ask for a Haussmann style, or Victorian style, by just providing an image with such buildings on it. The style vector can also be used to obtain results at dawn or during the winter. It also shed light on the remaining issues in this dataset. The misalignments between the target and the simulated image, especially when considering the building edges, are big. Unfortunately, there is no handy solution yet, except for helping Mapillary’s team develop a refined algorithm to find the exact camera coordinates and rotation vector. Six more examples of generated samples can be found in Figure A5.

### 3.2. Comparison against Other Models

This well-working SPADE generator was compared with other existing models, with both metric evaluations and human evaluation. The competing models are a CycleGAN with its two spade generators having a lower base filter number to fit in memory; a seven-level deep U-net; and finally, the same SPADE generator but optimised with hinge loss. All generated samples can be seen in Appendix A.

The SSIM scores presented in Table 1 put the CycleGAN first, well ahead of the other models. However, the visual inspection of Figure 10 shows that, if the CycleGAN outputs a refined representation of the building on the left, it totally misses out on the building on the right. The high score of that model can be explained by its mechanism, which makes the generated image resemble the target a lot. On the other hand, the SPADE that uses the hinge loss fails to achieve realistic outputs but does capture the sky along with greenery. The model with the lowest score is actually the one that produces the most realistic samples. It is likely that the *SPADE + L1 loss* model found a balance between resembling the target image to lower the L1 loss and generating realistic images to fool the discriminator. It highlights the difficulty of finding the right metric to evaluate a model.

For all the models, the generator was left untouched. However, the differences in its performance between the variations are notable as shown in Table 2. The weaker discriminator that focuses on distinguishing real street views from fake ones is once again CycleGAN. However, the contrast in the SPADE + hinge loss’s discriminator prediction is particularly impressive, leading to the conclusion that the discriminator was not challenged enough but the generator was. In Figure 11b–d, we can distinguish the sky taking a special place in the prediction.

### 3.3. Ablated Model

An ablation study was conducted to underline the role and influence of having extra depth and normal information in the inputs. Training has been performed by feeding the SPADE generator only the “DiffCol” pass, which represents a basic segmentation map. To accommodate the important decrease in input data, from 7 channels to just 3, the number of base filters has been reduced to 64. However, the number of upsampling layers was kept the same. This ablated model was trained on 265 epochs, with a batch size of 1.

The losses of the generator and the discriminator of this ablated model drawn in Figure 12 seem to tell two different stories. On the one hand, the adversarial loss (Figure 12a) increases, particularly after epoch 125. On the other hand, the discriminator validation loss on generated samples (Figure 12b) increases a lot after the 125th epoch, suggesting that the discriminator is not able to flag a generated sample as such.

The produced outputs of this ablated model are not realistic and are presented in Figure 13. Throughout the epoch, the network did not generate consistent results. Each new epoch had a totally different “style”. However, the sky was often well drawn with clouds and a light blue colour.

## 4. Discussion and Future Work

As seen in Section 3.1, the generated samples are very convincing, with proper building contours, facades, and a sky with a few clouds. However, the already-photorealistic results could certainly be improved. The samples were not all perfectly aligned, and some of them were outright irrelevant as they were not showing the city. Furthermore, the discriminator was maybe a bit too weak, and due to the different contexts present on a single image (sky, buildings, and streets), a context-aware discriminator as in [19] could be implemented to have a more efficient backward pass.

### 4.1. New Loading Mechanism

One of the bottlenecks of the training pipeline was the decoding of the image files that were saved in binary format, or PNG. For every new epoch, every sample needed to be decoded again, which impaired the performance for low batch size values. Considering the use of billions of images to train new diffusion models, it can be assumed that better techniques exist to load and augment images efficiently.

### 4.2. Better Discrimination

Considering the challenges faced during this study, the results are really promising. Nevertheless, it was noticed that on the SPADE generator, the discriminator was not strong enough against the generator. It could benefit from hyperparameter tuning, mainly to achieve faster training.

### 4.3. Video Stability

This model was never intended to be as video stable as the GANcraft (which also uses a SPADE network). However, considering it can be used in the automotive industry, it could be interesting to make the generated samples stable in the time dimension. It would allow the creation of seamless videos, which would be more useful to train autonomous driving models.

### 4.4. An Even Better Dataset

A few samples present in this dataset are not relevant enough to be seen by the model during training. For instance, there is one sample in front of an ATM where the 3D representation does not match the scene at all. A manual sorting beforehand could discard such samples to ease the training process and avoid artefacts in generated images.

The lack of metrics in the dataset makes the sample’s quality evaluation complicated. For instance, a metric could rely on the DiffCol pass, which gives enough context to explain how well a simulated sample is aligned with its matching street view. Due to the inherent characteristics’ variations between the two, real street views may benefit from preprocessing to smooth out plane surfaces and highlight the edges, as they contain more information about alignment. This quality evaluation could be used to remove the worst samples to improve the overall quality of the dataset.

Finally, in the 3D model of Paris used to generate the simulated images, a few buildings were missing, as shown in Figure A4. A new 3D data source could improve the generation of samples by having better fitted simulated samples and maybe even more details, such as the 3D roof’s shape, as in Google Earth, or trees.

## 5. Conclusions

The creation of a custom dataset was a huge task on its own, and the dataset has undergone a lot of improvements with the project moving forward. In its latest version, it simulates samples in their full numerical range (using 16 bits floats) that are aligned with their street view counterparts. The simulated domain comes in seven channels to represent depth, surface normal, and a segmentation map. As highlighted by the ablated model, this additional data are hugely beneficial for the generator network compared to regular RGB images. The trained GAN successfully enables the effortless generation of realistic images of street views. The results obtained with the SPADE generator are quite promising, and such a model could suit the automotive industry. Autonomous driving AI could be trained on samples generated by this model representing specific road configurations that are too rare in real life to be properly trained on.

Despite the good results, multiple solutions to improve the model’s stability and produce style-oriented results were proposed. These solutions could allow the model to produce non-human-discernibly generated samples.

The code is open source, documented, and under a creative commons licence inherited from the SPADE project. https://github.com/MarcBresson/Individual-Research-Project, accessed on 18 October 2023.

## Figures and Tables

**Figure 1 sensors-24-00094-f001:**
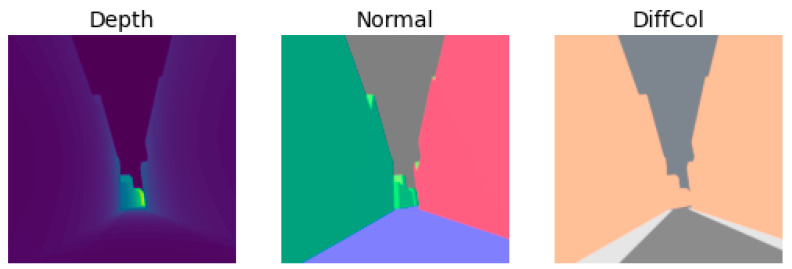
The three passes used for this work. The depth gives a sense of scale, the normal indicates an orientation, and the diffused colour segments the image into types of surfaces. These 3 images are cut from a larger 360° image.

**Figure 2 sensors-24-00094-f002:**
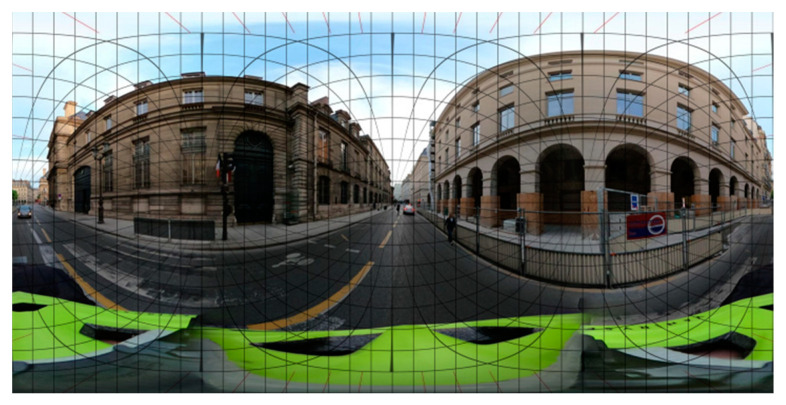
360° equirectangular image with geodesics superimposed. Image from Mapillary.

**Figure 3 sensors-24-00094-f003:**
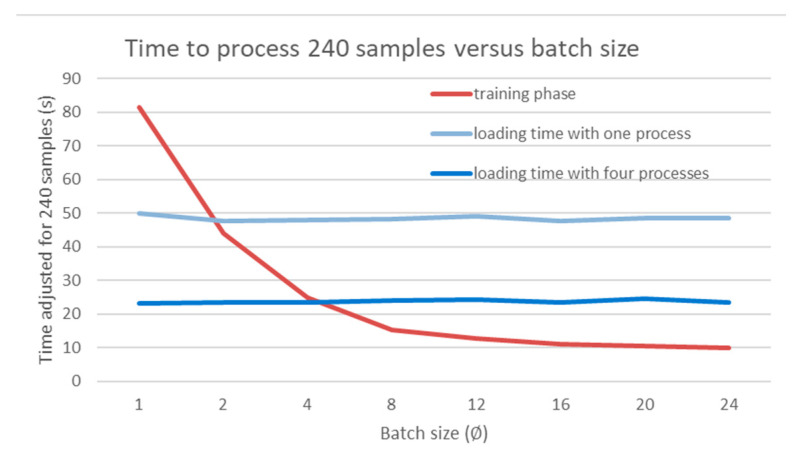
Time to process 240 samples against the batch size. The test was conducted with a single loading process and four loading processes.

**Figure 4 sensors-24-00094-f004:**
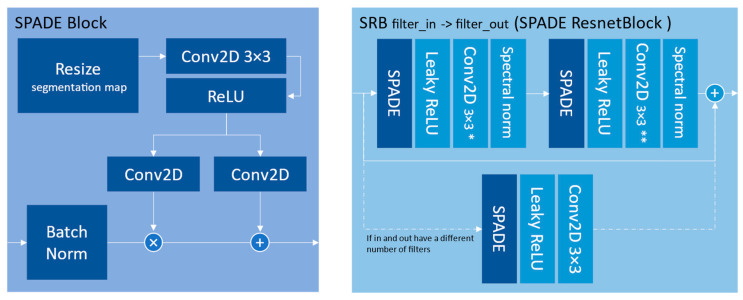
Architecture of the SPADE block and the SPADE ResNet block. * takes *filter_in* filters in input and outputs *min(filter_in, filter_out)* filters. ** takes *min(filter_in, filter_out)* filters in input and outputs *filter_out* filter.

**Figure 5 sensors-24-00094-f005:**
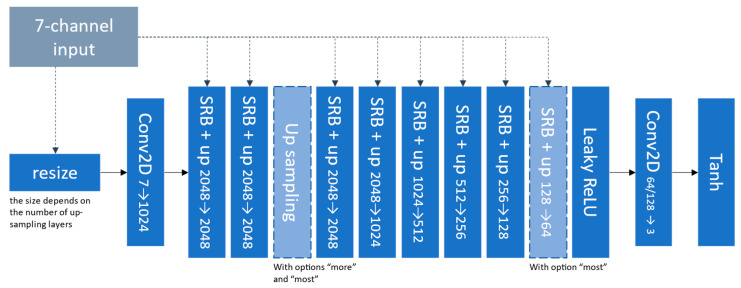
Architecture of the SPADE generator used in this work.

**Figure 6 sensors-24-00094-f006:**
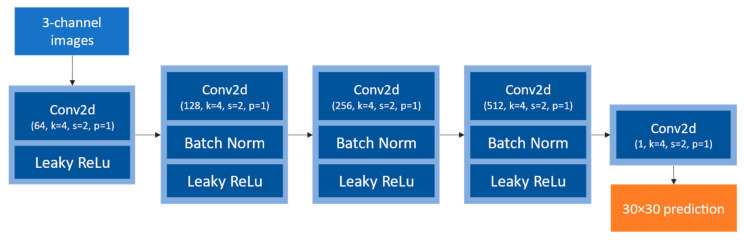
PatchGAN discriminator architecture. It outputs 30×30 px maps in the case of 256×256 px inputs.

**Figure 7 sensors-24-00094-f007:**
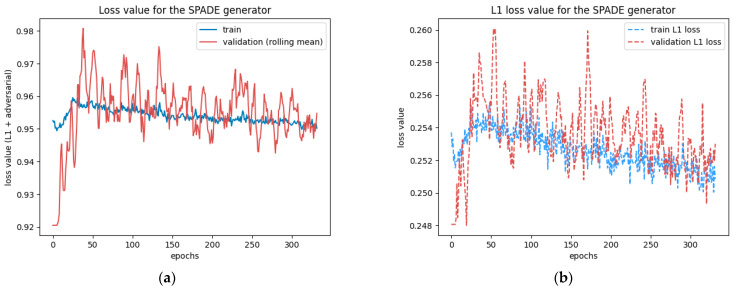
Loss values for the SPADE generator: (**a**) shows the total loss values combining the L1 loss and the adversarial loss, and (**b**) only shows the L1 loss values. They are represented on two different graphs because of their very different scales.

**Figure 8 sensors-24-00094-f008:**
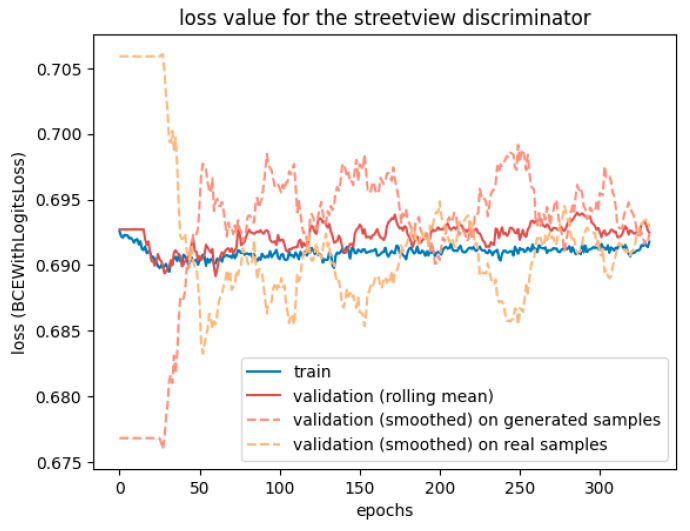
Loss values of the discriminator when trained with a SPADE generator. The three validation curves were smoothed out over 20 samples to represent the situation more clearly, hence the plateau in the first 20 epochs.

**Figure 9 sensors-24-00094-f009:**
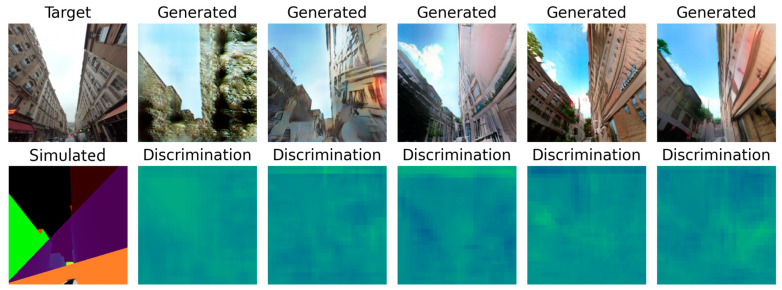
Generated samples with a SPADE generator at epochs 1, 20, 100, 330, and 331.

**Figure 10 sensors-24-00094-f010:**
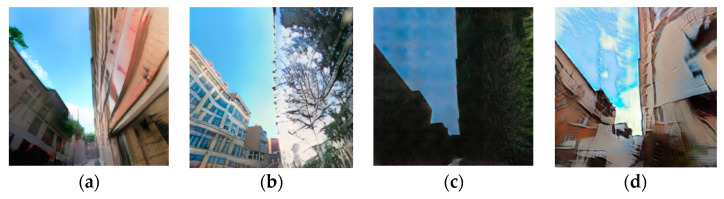
Last generated samples for the (**a**) SPADE + L1 generator, (**b**) the CycleGAN, (**c**) the U-net generator, and (**d**) the SPADE + hinge generator.

**Figure 11 sensors-24-00094-f011:**
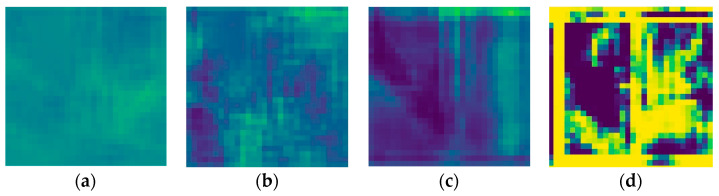
Last generated discrimination for the (**a**) SPADE + L1 network, (**b**) the CycleGAN network, (**c**) the U-net network, and (**d**) the SPADE + hinge network.

**Figure 12 sensors-24-00094-f012:**
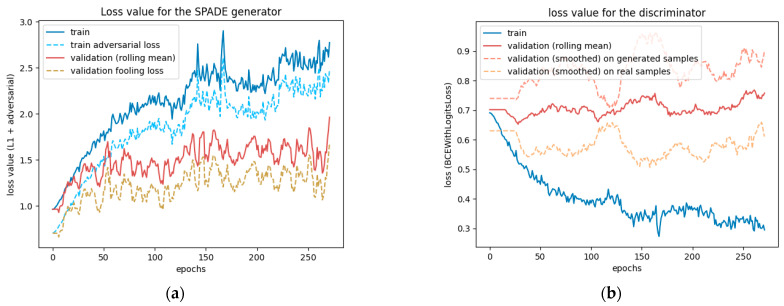
Loss values for the ablated SPADE generator and for the PatchGAN discriminator.

**Figure 13 sensors-24-00094-f013:**
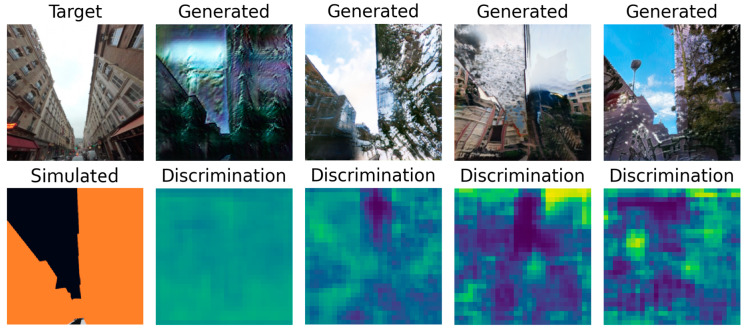
Generated sample with the ablated model at epochs 1, 20, 100, and 265.

**Table 1 sensors-24-00094-t001:** SSIM scores for tested models.

SPAPE + L1 Loss	CycleGAN + L1 Loss	U-Net + L1 Loss	SPADE + Hinge Loss
0.2860	0.8327	0.3584	0.3208

**Table 2 sensors-24-00094-t002:** Binary cross-entropy scores for the tested model’s discriminators.

SPAPE + L1 Loss	CycleGAN + L1 Loss	U-Net + L1 Loss	SPADE + Hinge Loss
0.6952	0.7159	0.5185	0.5045

## Data Availability

The “paired street view and simulated view in Paris” dataset created and used in this project is available freely at https://www.kaggle.com/datasets/MarcBresson/paired-street-view-and-simulated-view-in-paris accessed on 20 October 2023. All the code and methodology are described to generate more pairs in any other city.

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
