# Peer review of "Sim2Real: Generative AI to Enhance Photorealism through Domain Transfer with GAN and Seven-Chanel-360°-Paired-Images Dataset"

_sensors, 2023, doi:10.3390/s24010094_

Round 1
Reviewer 1 Report
Comments and Suggestions for Authors
The abstract needs to be restructured with the background problem addressed by the article and the solution developed, the result achieved and concluding remarks
The 7-channel images section is not clear; how 7 channels are related and utilised is missing.
Material and method sections are poorly articulated. need to restructure and present well with the help of a flowchart and flow diagram with clear flow and coherence.
The high value of the structural similarity index measure indicates more resembles or close match but under Table 1 and Table 2 author claims that less structural similarity index measure value shows more resembles. The claim needs to be checked.(Figure C has more value than Figure A but A has more resemblance to ref image?)
The author presented only 2 sample results in the article. The result section is to be enhanced with more analysis for example structure similarity index measurement result should be analysed with more data set with different category and the average performance of similarity measurement to be presented to understand the capability of the proposed model to generate an image set of different category.
Author Response
Hello, thank you very much for your comments. They were very valuable and I discussed them all below.
1) I reworded the abstract to follow the structure you suggested:
- background problem: data scarcity and the limited info contained in RGB to generate new images
- solution: created a GAN and used high-dimensionality images to have more data (thus control) over the generated samples
- results: good and photorealistic
- concluding remarks: generated samples could totally be used to train new AI
2) I reworded the 7-channel section to (hopefully) make it clearer.
3) The "Material and method" section is now organized like in a pipeline
- Data
- Models
- Evaluation
4) You are totally right. I got hyped up when I saw lower number for the most performant model from my human eye. Actually, this table only reveals that the SSIM metric is not suited for such a task. I changed the analysis of this part. Thank you very much for that remark.
5) I actually lost my student account because my scholarship ended between the submission of my article and now. I can't access the supercomputer any more so I added six new samples that I had already saved on my laptop. It is all under Appendix C as I could not provide further analysis from different categories.
Reviewer 2 Report
Comments and Suggestions for Authors
Image-based semantic scene segmentation is one of the most important tasks in computer vision. Although we have seen great progress in recent years with sophisticated image descriptors and more advanced machine learning techniques, segmentation is still a challenging task. While humans have no difficulty in performing semantic interpretation of images, machine vision systems still struggle. The problem of data scarcity when creating new images using generative-adversarial networks is a challenging task. To solve it, the authors use Generative Adversarial Networks for domain transfer, efficiently converting a multi-channel image from a 3D scene into a photorealistic image. This is accomplished using the authors' set of software findings, as well as a suggestion to use a certain pairwise arrangement of virtual cameras. The pre-training of the dataset for this transformation is utilized. The results obtained with the developed generator look promising. Solutions to improve the stability of the model are proposed. These solutions may allow the model to produce apparently realistic images. The material of the paper is presented quite clearly, including structure, a set of figures, and a list of references. The material is interesting enough and can be accepted for publication.
The problem of data scarcity when creating new images using generative-adversarial networks is a central task that the research address to.
The authors show how to use Generative Adversarial Networks for domain transfer, efficiently converting a multi-channel image from a 3D scene into a photorealistic image.
The main findings are accomplished using the authors' set of software findings, as well as a suggestion to use a certain pairwise arrangement of virtual cameras. The pre-training of the dataset for this transformation is utilized. The results obtained with the developed generator look promising. Solutions to improve the stability of the model are proposed. These solutions may allow the model to produce apparently realistic images. The authors show that their software solutions provide better results than existing ones in solving this problem.
Author Response
Hello, thank you very much for your nice detailed comments. They were much appreciated :)
Through the inputs of other reviewers, I reworked the article to make it even better and clearer. Notably, I had made one big mistake about the SSIM score because I implied that lower was better as the most realistic model also had the lowest score. However, the higher the SSIM the better, which meant that the realistic model was actually the one that made the furthest predictions from the target!
Thank you again for your comments and have a great day.
Reviewer 3 Report
Comments and Suggestions for Authors
This paper aims to provide a solution to data scarcity by allowing end users to create 8 new images while carefully controlling building shapes and surroundings.
The paper is not well-written and organized. The language and presentation of the paper should be thoroughly reviewed. In some parts of the paper, there are expressions such as "in this thesis" and in other places, "in this project". This study, which is ultimately evaluated as a journal paper candidate, should have a consistent and appropriate address throughout the paper.
The introduction is not clear and does not provide sufficient background to the problem at hand. What are realistic applications where end users experience data scarcity? Can you give concrete examples by making references to recent literature? The paper should effectively justify the need to improve photorealism through domain transfer with a custom 7-channel 360° mapped image dataset.
The last paragraph of the Introduction section should mention the organization of the rest of the article.
The literature review on the use of GANs for photorealism enhancement and domain transfer is not comprehensive and up-to-date. Recent studies (including 2021-2024 studies) should be included.
​The methodology is not well defined and easy to understand. Details of the custom 7-channel 360° mapped image dataset should be clearly stated.
What are any changes or innovations in GAN architecture that have contributed to the development of photorealism?
How to split your dataset into train, test, and validation sets? Did you use k cross validation?
What are the challenges or problems encountered during the experiments that need to be highlighted?
The computational burden of the algorithm/proposed model is not discussed. Please include a few sentences about the computational burden. Further clarification on the approach and methodology used in this context is necessary.
Implications of the findings, including potential applications and limitations, should be discussed?
It is unclear how the proposed method contributes to existing knowledge in the field.
The conclusion does not effectively summarize the main contributions and findings of the paper. The conclusion section should be improved.
What suggestions do you have for future research to consider?
Comments on the Quality of English LanguageDear Editor,
This paper aims to provide a solution to data scarcity by allowing end users to create 8 new images while carefully controlling building shapes and surroundings. Additionally, the corresponding author has published part of the paper on “github.com”. Does this pose a problem in terms of journal policy? My evaluations and concerns about the paper are listed in the comments and Suggestions for Authors section.
BR
Assoc. Porf. Dr. MHBilgehan UCAR
Author Response
Thank you very much for your honest review. I really appreciate your comments as they are the only thing that can make my article great and easily understandable. I took all of them into account and listed all the modifications brought to the article below.
The multiple typos stating "in this thesis" have been removed, and the use of "in this project" has been reduced to its strict minimum.
Added a paragraph to mention the loading burden. Added a few sentences to speak about inference time.
Application examples were added based on the original use case this work was done around (military use case), and also from the datasets that were considered earlier in this work that were built for autonomous driving. A source highlighting the issue with road geometry in autonomous driving has been added.
The plan of the article was added.
New sources (3 from 2023, 1 from 2022 and 1 from 2021) were reviewed and added in the literature review or in the introduction.
Further information were added on custom 7-channel 360° mapped image dataset in the introduction and in the corpus. A dedicated paragraph details what s in the dataset.
When the SPADE and PatchGAN architectures and are discussed, explanation for why it can achieve photorealism has been added.
A new (short) paragraph was added to address the splitting strategy.
Discussions around the challenges coming from the use of custom datasets were added in the "Mapillary" and "Perspective Transformation" chapters.
Added a paragraph under "discussion" to mention the loading burden. Added a few sentences to mention inference time in the second paragraph of the "results" section.
added a paragraph under "discussion" to mention limitations. Example of applications were given in the introduction and conclusion.
The conclusion has been revamped to highlights the contribution of the proposed method.
"future works" have been merged with the "discussion" chapter. A section about video stability was added.
Reviewer 4 Report
Comments and Suggestions for Authors
Sim2Real: Generative AI to Enhance Photorealism Through 2 Domain Transfer with GAN and 7-Chanel-360°-Paired-Images 3 Dataset
Overall, this work presents an innovative approach to image generation by tackling data scarcity through a sophisticated GAN network and a carefully curated dataset. The emphasis on preserving semantic information and achieving photorealistic results adds substantial value to the field, with potential implications for various applications requiring image generation in constrained data environments.
Comments
Consider refining the flow of ideas. Start with a clear introduction that provides an overview of the study's objectives and methodology before diving into specific details.
Ensure consistency and clarity in citing references. For instance, use consistent formatting for citations (e.g., [1], [2].
Overall, the document holds promise in exploring GANs for domain transfer using a specialized dataset. Enhancing the clarity of technical details, organizing the content for better readability, and offering more context to readers unfamiliar with certain concepts will strengthen the document's impact and accessibility.
The present work is suitable for publication after minor corrections.
Comments on the Quality of English LanguageMinor editing of English language required
Author Response
Hello and thank you very much for your positive feedback. I reworked my article quite a lot.
I rewrote the first part of the introduction as you suggested. It now provides an overview of the article along with its plan.
I went through all the citations to make sure I use the same formatting (I stick to [1][2][3] because that was what I used the most). However, I am not sure it was what you meant by "Ensure consistency and clarity in citing references". Do not hesitate to write back (either through this platform if it allows it, or directly via email at marco.bresson@gmail.com).
Round 2
Reviewer 3 Report
Comments and Suggestions for Authors
The authors have progressed in improving the paper compared to previous versions of the paper (2022-2707655_R1 & 2023-2707655_R2). When the previous and revised versions of the paper are evaluated together, it is seen that the authors make the corrections requested by the referees and show the necessary sensitivity in the revision of the paper in line with the comments. In the revised version of the paper, almost all the comments have been considered and addressed by the authors.
However, the authors did not prepare any "responses to reviewers" file. Therefore, the changes made by the authors in line with the opinions/suggestions/evaluations of the reviewers cannot be tracked.
The existing organization and spelling problems in the previous version of the article have been fixed. In the revised version, the clarity and follow-up of the study have been increased. In addition, the article has been carefully reviewed for grammatical and typos.
All in all, my concerns on the previous version of the paper have disappeared with the explanations made by the authors, as well as the revision they have made.
This revision is sufficient, and it is possible to evaluate the paper for publication after necessary checks for minor spelling errors and grammar check.